# Promoting Sports Practice in Persons with Hemophilia: A Survey of Clinicians’ Perspective

**DOI:** 10.3390/ijerph182211841

**Published:** 2021-11-11

**Authors:** Giuseppe Lassandro, Domenico Accettura, Paola Giordano

**Affiliations:** 1Department of Biomedical Science and Human Oncology—Pediatric Unit, University of Bari “Aldo Moro”, 70121 Bari, Italy; giuseppelassandro@live.com; 2Istituto di Medicina dello Sport-FMSI, 70132 Bari, Italy; presidente@medicidellosport.it

**Keywords:** hemophilia, sports, quality of life, bleeding, survey

## Abstract

Historically, people with hemophilia have been warned to avoid physical activities as a possible cause of bleeding; however, currently, sport is considered necessary, especially in the developmental age, for providing a good quality of life. A survey was proposed to a group of hematologists and sports physicians working in Puglia, Italy, to explore their approach to physical activities for their patients with hemophilia and to obtain suggestions about possible interventions to promote the access of patients to sports. The survey was answered by 6 hematologists and 15 sports physicians. In total, 71% (about six patients/year/physician) of patients with hemophilia seen by sports physicians asked for counseling about sports, and 67% (about five patients/year/physician) actually practiced sports. On the other hand, only 31% (about 16 patients/year/hematologist) of patients asked hematologists questions on sports, and only 16% (about seven patients/year/hematologist) of patients with hemophilia and that were followed-up by hematologists practiced sports. The sports most often recommended to patients with hemophilia by physicians included swimming, athletics, tennis, running and gymnastics. According to hematologists, physical activity was very efficient in improving the quality of life of patients; stability of joints; their psychological, social and musculoskeletal wellbeing; and in reducing the risk of bleedings. On the other hand, physical activity was considered less important in all these areas by sport physicians. In conclusion, answers to this survey suggested that sports could be promoted among hemophilic patients by increasing the sports physicians’ knowledge about hemophilia and their special role in this area. In addition, interviewed clinicians were of the opinion that increased awareness of specific guidelines and clinical practice protocols among both hematologists and sports physicians could be beneficial. Finally, answers suggested that access to fitness certification should be facilitated.

## 1. Introduction

Hemophilia A and B are the most common X-linked inherited bleeding disorders resulting, respectively, from factor VIII and factor IX protein deficiency or dysfunction. They are characterized by prolonged and excessive repeated bleedings after minor trauma or sometimes even spontaneously. The disease is considered mild, moderate or severe on the basis of the factor levels in basal conditions [1]. Patients with mild disease (5–40% of normal level of factor VIII or IX) usually do not have spontaneous bleedings but bleed after trauma. When the factor level is 1–5% of the norm, hemophilia is moderate and bleeding follows after trauma, surgery or dental procedure [2]. Joint bleeding is characteristic of hemophilia, and repeated episodes progressively result in arthropathy and movement impairment up to disability. Repeated knee joint bleeding was found to be responsible for quadriceps atrophy and strength deficit in patients with severe and moderate hemophilia [3].

The introduction of factor VIII and IX replacement therapy reduced the impact of bleedings, and the introduction of prophylaxis helped to decrease the frequency of bleedings and slowed the progression of joint disease [4]. In addition, factor prophylaxis helped people with hemophilia to move freely, with little fear of trauma, and favoured a better quality of life [4,5]. Different replacement therapy regimens are currently available, and patients with hemophilia A and B may receive either prophylaxis or on-demand treatment by using different types of clotting factor concentrates [6,7]. The opportunity of an efficient prophylaxis reduces the risk of bleedings and should also reduce patients’ and caregivers’ fear of the noxious consequences of physical activity.

Historically, prior to the introduction of factor concentrates, prophylaxis for bleeding in patients with hemophilia was represented by the avoidance of activities that may result in joint trauma. The introduction of prophylaxis with factor concentrates reduced the risk of bleedings due to trauma and made exercise accessible to people with hemophilia, but also in recent years, patients have been requested to limit many types of activities, such as contact sports, which were considered to possibly result in severe bleeding events and to increase the risk of progression to arthropathy [8]. Nevertheless, as quality of life has become the main objective of patients’ management, physical activity and exercise are considered necessary for people with hemophilia, especially for patients in the developmental age. Moreover, it has been observed that exercise may help strengthen muscles, maintain joint mobility and bone health, improve physical functioning and reduce the impact of bone aging; all of these effects are useful for maximizing the healthy efficiency of treatments for hemophilia [8,9,10]. In addition, several authors believe that physical activity and exercise are beneficial for specific outcomes of patients with either hemophilia type A or B, such as reducing bleedings, and that they can improve the quality of life [11,12,13,14,15,16,17].

It may also be mentioned that reduced exercise places people with hemophilia (PWH) at increased risk to becoming overweight or obese; indeed, a higher mean body fat rate was measured in patients with hemophilia compared to controls (23.5 vs. 17.0%, *p* = 0.055) [18]. Such findings suggest that the avoidance of exercise may place people with hemophilia at increased risk of osteoarthrosis and of cardiovascular events [19,20]. Indeed, a recent study showed that PWH had a higher prevalence of blood hypertension compared to the general population in Brazil, and about two-fifths have a high risk of developing a cardiovascular event in the following 10 years [21]. Obesity and hypertension were frequent in a group of 48 PWH with a mean age of 21 ± 9 years, especially in those with arthropathy [22]. Weight control may, thus, be beneficial to the general health and to joints; in addition, it may help to limit factor dosing, which is based on body weight.

Of course, exercise is associated with specific challenges for subjects with hemophilia and must be correctly managed. Little is known about the opinion of physicians on the opportunity to promote exercise and sports in these types of patients and their approach to this issue in clinical practice.

The aim of this study was to investigate the opinion of physicians concerned with the management of patients with hemophilia toward the practice of sports and their experience with patients, with a focus on the barriers to sports practice in a limited setting.

A survey was proposed to a group of hematologists and sports physicians working in a localized area (Puglia, Italy) in order to explore their approach to sports for their patients with hemophilia and to obtain suggestions about possible interventions to promote the access of patients to sports.

## 2. Methods

A survey was developed with the assistance of an independent third party, with broad experience in qualitative and quantitative research in the pharmaceutical setting (DoxaPharma, Milan, Italy). The questionnaire was delivered online via a computer-assisted web interview between January and February 2021. The questionnaire contained 29 questions. Open and closed (multiple-choice questions, with either single or multiple permitted answers) questions were included (Appendix A). Interviews were anonymous. The level of agreement was measured by a 5-point Likert scale. Data were analyzed by descriptive statistics.

The survey was performed in Puglia, an Italian region. This area is culturally homogeneous and has a regional health system, providing similar assistance to people living in the region. The respondents work in similar conditions and may resort to similar resources. All hematologists working in Hemophilia centers in Puglia (Italy) and a group of sports physicians were invited to participate. Sports physicians from all over Puglia were selected by the Italian Sports Physicians Association to represent the entire area.

The release of a certificate of fitness is required to practice sports in Italy. It is released by specialists after a visit, a person with hemophilia (PWH) and any other investigation as considered necessary. Possible problems of patients with hemophilia in obtaining this certificate are investigated in the perspective of Italian regulation.

## 3. Results

### 3.1. The Sample

The survey was answered by 6 hematologists and 15 sports physicians. The responding hematologists reported visiting an average of 675 patients per year; 7% (50/675) of their patients were affected with hemophilia type A or B. On average, sports physicians reported assisting 1033 patients per year; only 0.8% (*n* = 8) of these subjects were affected with hemophilia. Overall, responses were related to 120 patients for sports physicians and to 300 patients for hematologists. Hemophilia type A represents 63% (53/120) of cases of hemophilia observed by sports physicians, and 72% (38/53) of them were reported to be followed up by hematologists. Almost 50% (115) of patients with hemophilia were older than 18 years. Each patient with hemophilia was found to be visited in an average of three times per year by a sports physician and 11 times per year by a hematologist. The hematologists reported that counseling the patients regarding physical activity accounted for 13% of the time spent for each visit. 

### 3.2. Current Management of Sports in Hemophilic Patients

In total, 71% of patients (about 6 patients/year/physician) with hemophilia seen by sports physicians ask for counseling about sports, and 67% (about 5 patients/year/physician) practice sports (20% of this group practice competitive sports). On the other side, only 31 of patients % (about 16 patients/year/hematologist) ask hematologists questions on sports and only 16% (about 7 patients/year/hematologist) of patients with hemophilia and followed-up by hematologists practice sports (only 2% of them practice competitive sport).

Sports most often recommended to patients with hemophilia by sports physicians included swimming (93% of physicians), athletics (33%), tennis (27%), running (27%) and gymnastics (27%). The hematologists most often recommend swimming (67% of hematologists), cycling (67%), athletics (33%) and football (33%).

Both hematologists and sports physicians recommended avoiding some sports, with some differences in the frequency of the recommendation: boxing/fighting (83 and 93%, respectively, for hematologists and sports physicians), football (33 and 67%, respectively), basketball (33 and 40%, respectively) and rugby (33% for both groups of physicians).

Only 50% (*n* = 3) of hematologists and 7% (*n* = 1) of sports physicians were aware that national or international guidelines for the physical activity of patients with hemophilia were available.

In addition, 17% (*n* = 1) of hematologists and 27% (*n* = 4) of sports physicians reported that a protocol for the management of sports practice in hemophilic subjects was available in their center.

Participants’ opinions on the relevance of a regular physical activity (exercise) in hemophilic patients are reported in Table 1. Briefly, according to the hematologists, regular physical activity was very important for improving the life of patients; stability of joints; their psychological, social and musculoskeletal wellbeing; and for reducing the risk of bleedings. On the other hand, regular physical activity was considered less important in all these areas by sports physicians.

All hematologists answered that it is very important that children and adolescents with hemophilia regularly practice sports in order to improve joint functionality and bone density and to reduce the risk of bleeding. Only 40% (*n* = 6) of sports physicians shared this opinion.

Both hematologists and sports physicians reported that the main barriers to practicing sports for patients with hemophilia are the patient’s fear of bleeding induced by trauma and the lack of correct information about opportunities (Figure 1). Parents’ and relatives’ objections are also an important obstacle. The wish to practice an inadequate sport is considered an obstacle by 53% (*n* = 8) of sports physicians, while this problem is considered only by 17% (*n* = 1) of hematologists. Finally, 67% (*n* = 4) of hematologists and only 13% (*n* = 2) of sports physicians answered that hemophilic patients find it difficult to obtain a certificate of fitness. Fitness is usually certified in 13 days, after two visits, and is checked twice a year. 

Patients are referred to other specialists before the release of the certificate in 80% of cases if a competitive sport is concerned and 47% of cases if an amateur sport is practiced. The other specialist is most often the hematologist.

### 3.3. Possible Strategies for Improvement

The hematologists suggested that the practice of sports could be better promoted in hemophilic patients if the compliance to the prophylaxis was increased (100% of hematologists, *n* = 6) and factor was infused before the activity (according to 67%, *n* = 4). According to sports physicians, these interventions would not be relevant, while patients could benefit from greater availability of emergency resources (i.e., a free phone number of the hematology center; training in cases of emergency for coaches).

Only 50% (*n* = 3) of hematologists and 60% (*n* = 8) of sports physicians agreed that fitness certificates should be released upon documentation of joint examination and good compliance to therapy. In addition, 83% (*n* = 5) of hematologists and only 53% (*n* = 7) of sports physicians agreed that the hematologist should personalize therapy when the patient initiates practicing sports, according to the clinical condition and to suggestions from the sports physician. A total of 67% (*n* = 4) of hematologists and only 33% (*n* = 5) of sports physicians agreed that hematologists could suggest practicing amateur exercise while consulting the sports physician. Only 67% (*n* = 4) of hematologists and 20% (*n* = 3) of sports physicians agreed that patients with hemophilia wanting to practice sports should be referred to a dedicated office for fitness certificate release. Only 13% of sports physicians and 67% (*n* = 4) of hematologists agreed that fitness certificates for patients with hemophilia should be granted by a multidisciplinary team including the sports physician, the hematologist and the physiatrist. Only 50% (*n* = 3) of hematologists and 13% (*n* = 2) of sports physicians agreed that telemedicine could help the release of a fitness certificate to patients with hemophilia. 

## 4. Discussion

The survey described here investigated the opinion of specialists on the practice of sports by subjects with hemophilia, their awareness about specific guidelines and protocols and asked about possible interventions to promote sports among patients. This survey was performed in a limited area that was culturally homogeneous in Italy and identified specific needs, suggesting possible interventions. This approach was chosen to obtain information that could be easily and directly applied in clinical practices.

Our results confirmed the hypothesis that a low proportion of patients practice sports. Overall, hematologists attributed a greater relevance to sports for hemophilic patients in comparison with sports physicians. The latter ones were rarely aware of recommendations on the issue, and only a low proportion of them agreed that sports may improve the joint functionality of patients.

In the opinion of respondents, the main reasons why patients rarely practice sports were the fear of trauma, little information about opportunities and relatives being fearful of bleeding. Hematologists but not sports physicians reported that hemophilic patients find it difficult to obtain their fitness certificate to be admitted to sports centers. Thus, it is not only organization improvements that could be beneficial but increased awareness of sports physicians is also necessary.

Our results suggest that patients with hemophilia should practice sports under the surveillance of specialists, that expert counselling for the choice of appropriate sports should be available and that the sports practice could be considered safer by patients and caregivers if followed up by specialised physicians. In addition, it was found that prophylaxis is followed regularly by patients in the area of the survey, and this should help physicians to propose physical activity to their patients with little concern for trauma consequences.

A limitation of this study may be the small number of physicians (6 hematologists and 15 sports physicians) and of patients described by the respondents (120 patients followed-up by sports physicians and 300 patients by hematologists). The results may be considered preliminary and suggestive for further investigations.

Studies have repeatedly demonstrated that exercise is useful to people with hemophilia with respect to improving joint health, movement and quality of life without increasing bleedings [9,10,11,12,13,14,15,16,17]. In a sample of 50 adults with hemophilia A, being more involved in sports was related with better health related quality of life (HRQoL) in comparison with being involved in sports for a lower period of time (*p* < 0.005). Regardless of hemophilia severity, sports require great attention, as this is one of the most impaired domains of quality of life (QoL) in children and adolescent patients [23]. Participation in sports activity was found to help PWH in terms of being a part of a group with an increased number of friends and with respect to increased self-esteem [2]. In addition, those that were more involved in physical activity had better physical performances [24]. Little exercise is linked to muscle weakness, instability and risk for stress of joints, with a progression to risk of joint lesions, pain and immobility [25]. Regular physical activity was observed to improve strength, proprioception and range of motion in patients with hemophilia, resulting in a prevention of the destructive processes in joints [13]. Although exercise is deemed important for patients with hemophilia in maintaining the functionality of joints and seems to help reduce the risk of bleeding, patients and their families are often reluctant to this practice, because they are afraid of trauma. This attitude used to be shared by caregivers, patients and physicians, but recent observations showed a changing situation. A recent observational study on adult patients with hemophilia in Great Britain found that 85% of participants met the UK physical activity guidelines, but joint disease and severity type influenced the amount of activity undertaken [26]. In the Netherlands, Timmer et al. found that the movement behaviour of adults with severe hemophilia was characterized by less walking and less running in comparison with healthy adults [27]. On the contrary, a nationwide, cross-sectional study in Dutch PWH suggested that sports participation in PWH was comparable to the general population [28]. Very little data existed on the activity levels of PWH in countries with limited resources [29]. In Italy, the attitude toward sports practice in PWH was discouraging in 2018 [30].

Our findings were in agreement with previous observations and with other authors’ studies, which demonstrated that the lack of knowledge and misunderstanding of the preventive effects of physical activity in hemophilia were among the main barriers to practicing sports [30,31,32]. The fear of parents was found to manifest with excessive stress in terms of the risks linked to non-recommended exercise activities [31]. An improved communication to patients and caregivers seems to be a mandatory aim of any attempt to promote the participation of PWH with respect to sports. 

A pivotal issue to promoting exercises of PWH is the identification of a suitable activity. Overall, exercise may be considered appropriate if it is adapted to the special needs of the individual person with hemophilia and recommended by an experienced physiotherapist or hemophilia comprehensive care team or described in the scientific literature [29]. Although recommendations are available, each patient should have a consultation and a full assessment with a trained professional, possibly taking part in a team, before starting an exercise program. In our study, interviewed sports physicians thought that boxing is to be avoided, in agreement with previously published opinions. A survey on physical therapists reported that risks for hemophilia patients inherent to a sport included impacts with surface/ball/equipment [33]. The choice of a suitable sport may require specific medical counseling. Counseling by specialists, mainly hematologists, will be efficient if physicians are aware of the current recommendations about practicing sports for hemophilia patients and if they feel confident in the organization of the healthcare system. In the last revision of guidelines, the National Hemophilia Foundation recommended that patients meet with their healthcare provider for a musculoskeletal evaluation before starting a new activity [11]. Factors to be considered in the choice of the sport include the patient’s interests, bleeding history, risks and benefits inherent to the activity and the current physical condition [11]. Physicians should pay increased attention to symptoms of patients practicing sports; bleeds may be not obvious in patients with mild disease. In addition, the history of bleeding frequency during exercise can be used as a tool to determine prophylaxis need [11]. A recent experience with counselling to young patients with hemophilia showed an increase in activity behaviour and physical fitness without increasing bleeding rate and maintaining joint function [34].

Patients, even children, may have functional deficits related to hemophilia and often exhibit subclinical joint findings; motion analysis and orthopedic examination should guide the choice of sport [35]. In order to prevent injury during sports practice, strength conditioning should be performed prior to progressing to higher level activities. Healthcare providers (including the hematologist, the sports physician and the coach) should agree with the person with hemophilia on a slow increase in activity over a pre-determined period of time [11]. Getting started is considered a hard step toward becoming physically active, and the National Hemophilia Foundation recommends the use of several resources that may help to overcome barriers, mainly by educating the patient on the benefits of physical activity [2,11]. 

A randomized controlled trial including 20 adult patients with hemophilia found that progressive strength training with elastic resistance performed twice a week during 8 weeks was safe and effective in improving muscle strength and functional capacity and reducing pain [36]. The patient should be followed up by the sports physician, and where this is not possible, online home-based training could be offered, as proposed by Wagner et al. [37]. Decreases in future injury risk and decreased joint destruction were obtained through several types of exercises, including postural training and coordination, swimming, cycling, martial arts, golf, walks and basketball [13]. A specific sports therapy focused on proprioceptive functions and accompanied by gentle strength training with low resistance and 20–25 repetitions was found to increase proprioceptive performance and muscular strength with minimal stress to joints in a clinical study [38].

In a model of hemophilia B mice, treadmill exercise resulted in a high incidence of muscle bleeds, but recombinant factor IX (rFIX) treatment before treadmill exercise prevented muscle bleeds, suggesting that appropriate prophylaxis increased safety during exercise [39]. Nevertheless, a recent study found that sports participation is not associated with adherence to prophylaxis in Dutch patients with hemophilia, suggesting that patients may not be aware of the importance of prophylaxis [40]. Finally, further investigation is needed to understand if patients would be available to follow physicians’ directions once an attentive assessment and a strict monitoring were offered. A better understanding of the relationship between physical activity and severity of the clinical disease, bleeding phenotype and treatment regimen (mainly, the frequency of administration may be relevant) is required. 

## 5. Conclusions

In conclusion, on the basis of the answers to this survey, some possible areas of interventions to promote sports among hemophilic patients could be identified. The answers suggested that awareness of the importance of physical activity for these patients is higher among hematologists than among sports physicians; thus, increased knowledge of hemophilia among sports physicians could be beneficial. In addition, hematologists were of the opinion that fitness certificate releases could be facilitated. Finally, answers showed that awareness of specific guidelines and clinical practice protocols among both hematologists and sports physicians was low.

## Figures and Tables

**Figure 1 ijerph-18-11841-f001:**
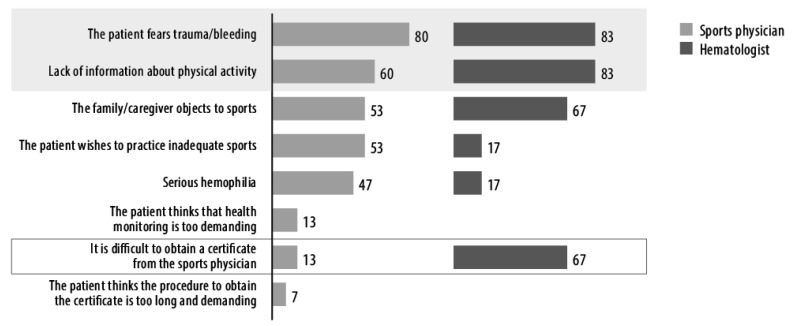
Barriers to the practice of sports by patients with hemophilia according to sports physicians and hematologists. Data are presented as percentage of responders, either sports physicians (*n* = 15) or hematologists (*n* = 6).

**Table 1 ijerph-18-11841-t001:** The opinion of sports physicians and hematologists on the relevance of regular physical activity (exercise) in people with hemophilia; percentage of participants expressing a high level of agreement with the statement.

Statement	Sports Physicians (*n* = 15), %	Hematologists (*n* = 6), %
A regular physical activity improves the overall quality of life of the patient with hemophilia	67	83
It is very important that people with hemophilia practice a regular physical activity to improve their psychological, emotional and social wellbeing	60	100
I always encourage my patients with hemophilia to practice physical activity/sports regularly	53	67
It is very important that people with hemophilia regularly practice physical activity/sports for their musculoskeletal wellbeing	47	67
A regular physical activity helps the patient to improve the stability and functionality of the joints and reduces the risk of acute bleeding and complications	47	83
Only a few sports are suitable for people with hemophilia	47	17
Patients with hemophilia in the developmental age should regularly practice physical activity/sports to improve joint stability and functionality, bone density and to reduce the risk of bleeding	40	100
It is not easy to encourage patients to practice sports because parents are afraid	33	17
The physical activity/sport is not indicated for all patients with hemophilia	20	17

## Data Availability

Data are available from the corresponding author upon due request.

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
