# Peer review of "Promoting Sports Practice in Persons with Hemophilia: A Survey of Clinicians’ Perspective"

_ijerph, 2021, doi:10.3390/ijerph182211841_

Round 1

Reviewer 1 Report

The title is ambiguous and general especially the word “Issues” and does not denote the author's findings.

I would recommend adding line numbers for easy revising.

Line 7. Insert “c” in factor

Please explain further how physical activity (PA) can improve quality of life? If the authors refer to PA how many minutes? Intensity? VO2max level? Does it depend on age and gender?

Please define PWH before using it.

It is known that when people do not perform PA in the form of exercise affections such as obesity and cardiovascular diseases appear. Can the authors explain how this situation is different in persons with hemophilia? Can the authors cite more studies besides the one of Putz et al, please?

Authors state that “Due to various reasons, including the relative rarity of the disease and the complexity of symptoms, little evidence on the benefit of exercise is available.” This statement is so general, Which reasons? Complex symptoms? Please explain further and add the appropriate references.

Insert “a” in Health in the Methods section.

Why do the authors use such a limited statistic?

It is highly recommended to include the survey as Supplemental Material or within the text.

It is highly recommended to present the % explained in Results in tables or graphs.

What was the % of females and males in the sample?

Omit “t” in respecttively. Check the words within the text: Participants’, Parents’, relatives’

Omit ”p” in hempphilia

Define the term HRQoL before using it.

What does the author mean by “less sports”?

It is important that authors understand the difference between physical activity, exercise, and sports and do not use these terms indistinctly all over the text. Please see

  1. E. Garber, B. Blissmer, M. R. Deschenes, B. A. Franklin, M. J. Lamonte, I.-M. Lee, et al., "Quantity and Quality of Exercise for 645 Developing and Maintaining Cardiorespiratory, Musculoskeletal, and Neuromotor Fitness in Apparently Healthy Adults: 646 Guidance for Prescribing Exercise," Medicine & Science in Sports & Exercise, vol. 43, pp. 1334-1359, 2011.

Discussion is mixed with Results. Please edit as appropriate.

Check 28 Berube 2017 reference in the Discussion section.

Please check that References have the same style i.e. #2,#10,#27,#35  

Reviewer 2 Report

This is an interesting attempt to survey an important and emerging issue in hemophilia - the encouragement of safe activities. I appreciate the effort to gather the perspectives of hemophilia and sports medicine practitioners.

The major conclusion to me in reading the results is that sports medicine specialists have serious knowledge gaps in hemophilia that go beyond the benefits of physical activity spelled out in the conclusions. They do not understand the role of prophylaxis or the appropriate sport selections. The paper comments specifically that they are "rarely aware of recommendations". I would suggest that the conclusions be expanded to include this broader lack of awareness.

The sports certificate needs to be defined and described in the introduction - this is not something I am familiar with for my patients or in my setting. The attitudes of the sports medicine specialists towards these certificates for hemophilia patients also illustrates a bias towards their own specializations, which may be appropriate depending on the nature of these certificates.

The methods section should describe in more detail the response rate to the surveys. The overall number of patients included is very low, especially for the sports medicine specialists. Is this representative of the population, or a selection bias of respondents? It certainly suggests an imprecision of the sports medicine assessments. 

Reviewer 3 Report

The article of Lassandro G & al., reports through a survey the management skills of hematologists and exercise physiologist to promote exercise with their patients suffering from hemophilia. Lassandro G & al. contribution is significant and enrich the research literature in a field where few studies are available.

The survey was answered by exercise physiologists and hematologist consulting hemophilia patients. The authors tried to determine if the specialists promote access to sports or manage patients who counseling about sports.

In results section,  the number of patients counseling the sport physiologist is too low comparing to hematologists n=8 vs n=50 which is expected and show the low promotion of practicing a sport among Hemophilic patients.

A general remark: The data were analyzed by descriptive statistics, however, most results are expressed in percentage and especially with regard to the abstract, it is misleading. Instead of percentages, it would be better to moderate the expression in percentage.

A second remark, I am aware that there are not many scientific studies or data available particularly with Hemophilia patients following an exercise protocol or program, that might explain why the introduction is mainly based on review articles (r.g ref: 8-9-10 then ref: 11-17). However, it would be better to put few studies such as Hilberg T, 2003 or others or even one study in murine models exercise treatment with recombinant factor IX (Tranholm M, 2015) that could be added at the end of the discussion.
